# Towards Lightweight, Model-Agnostic and Diversity-Aware Active Anomaly Detection

**Xu Zhang**[1], **Yuan Zhao**[2], **Ziang Cui**[3], **Liqun Li**[1], **Shilin He**[1], **Qingwei Lin**[1],[*]
**Yingnong Dang**[4], **Saravan Rajmohan**[5], **Dongmei Zhang**[1]
[1]Microsoft Research, [2]Peking University, [3]Southeast University, [4]Microsoft Azure, [5]Microsoft 365

## Abstract

Active Anomaly Discovery (AAD) is flourishing in the anomaly detection research area, which aims to incorporate analysts' feedback into unsupervised anomaly detectors. However, existing AAD approaches usually prioritize the samples with the highest anomaly scores for user labeling, which hinders the exploration of anomalies that were initially ranked lower. Besides, most existing AAD approaches are specially tailored for a certain unsupervised detector, making it difficult to extend to other detection models. To tackle these problems, we propose a lightweight, model-agnostic and diversity-aware AAD method, named LMADA. In LMADA, we design a diversity-aware sample selector powered by Determinantal Point Process (DPP). It considers the diversity of samples in addition to their anomaly scores for feedback querying. Furthermore, we propose a model-agnostic tuner. It approximates diverse unsupervised detectors with a unified proxy model, based on which the feedback information is incorporated by a lightweight non-linear representation adjuster. Through extensive experiments on 8 public datasets, LMADA achieved 74% F1-Score improvement on average, outperforming other comparative AAD approaches. Besides, LMADA can also achieve significant performance boosting under any unsupervised detectors.

## 1 Introduction

Anomaly detection aims to detect the data samples that exhibit significantly different behaviors compared with the majority. It has been applied in various domains, such as fraud detection (John & Naaz, 2019), cyber intrusion detection (Sadaf & Sultana, 2020), medical diagnosis (Fernando et al., 2021), and incident detection (Wang et al., 2020). Numerous unsupervised anomaly detectors have been proposed (Zhao et al., 2019; Boukerche et al., 2020; Wang et al., 2019). However, practitioners are usually unsatisfied with their detection accuracy (Das et al., 2016), because there is usually a discrepancy between the detected outliers and the actual anomalies of interest to users (Das et al., 2017; Zha et al., 2020; Siddiqui et al., 2018). To mitigate this problem, Active Anomaly Discovery (AAD) (Das et al., 2016), is proposed to incorporate analyst's feedback into unsupervised detectors so that the detection output better matches the actual anomalies.

The general workflow of Active Anomaly Discovery is shown in Fig.1. In the beginning, a *base* unsupervised anomaly detector is initially trained. After that, a small number of samples are selected to present to analysts for querying feedback. The labeled samples are then utilized to update the detector for feedback information incorporation. Based on the updated detection model, a new set of samples are recommended for the next feedback iteration. Finally, the tuned detection model is ready to be applied after multiple feedback iterations, until the labeling budget is exhausted.

Despite the progress of existing AAD methods (Das et al., 2017; Zha et al., 2020; Siddiqui et al., 2018; Keller et al., 2012; Zhang et al., 2019; Li et al., 2019; Das et al., 2016), some intrinsic limitations of these approaches still pose great barriers to their real-world applications. Firstly, most AAD methods adopt the *top-selection strategy* for the feedback querying (Das et al., 2017; Zha et al., 2020; Siddiqui et al., 2018; Li et al., 2019), i.e., the samples with the highest anomaly scores are always prioritized for user labeling. However, it hinders exploring the actual anomalies that are not initially scored highly by the base detector. As such, these AAD approaches are

---

[*]Qingwei Lin is the corresponding author.

highly susceptible to over-fitting to the top-ranked samples, resulting in a suboptimal recall with respect to all anomalies. We shall demonstrate this with a real example in Sec. 2.1. Secondly, most existing AAD approaches (Das et al., 2017; 2016; Siddiqui et al., 2018) are tightly tailored for a certain kind of detection model, making it difficult to extend to other unsupervised detectors.

They need to modify the internal structure of a particular type of unsupervised detector, endowing them with the ability of feedback integration. Therefore, it is impractical and ad-hoc to re-design them each time facing such a variety of unsupervised detection models. Recent AAD methods (Zha et al., 2020; Li et al., 2019)

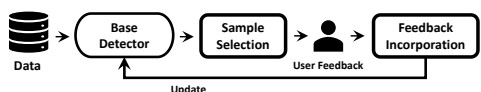

Figure 1: The general workflow of AAD.

attempted to generalize to arbitrary detectors. However, they can barely scale because their mode size grows with the number of samples.

To tackle these problems in AAD, we propose a **L**ightweight, **M**odel-**A**gnostic and **D**iversity-**A**ware active anomaly detection approach, named LMADA. It consists of two components, i.e, sample selector (for sample selection) and model tuner (for feedback incorporation). In the sample selector, we take the anomaly scores as well as the diversity of samples into account, instead of solely picking up the most anomalous ones for feedback querying. Specifically, we fuse anomaly scores and the feedback repulsion scores into a diversity-aware sampling technology powered by Determinantal Point Processes (DPP) (Chen et al., 2018; Kulesza et al., 2012). In the model tuner, we first leverage a neural network as the proxy model to approximate an arbitrary unsupervised detector. After that, we fix the weights of the proxy model and learn a representation adjuster on top of it. The representation adjuster is responsible for transforming the input feature vector to fit the feedback-labeled samples. Finally, each sample to be detected is transformed by the representation adjuster and then fed back to the base detector to estimate its anomaly score. In this way, the model tuner shields the details of different unsupervised detectors and achieves lightweight feedback incorporation, only via a non-linear representation transformation.

We conducted extensive experiments on 8 public AD datasets to evaluate the effectiveness of our proposed method. The experimental results show that LMADA can achieve 74% F1-Score improvement on average, outperforming other comparative AAD approaches under the same feedback sample budget. In addition, we also validated that LMADA works well under various unsupervised anomaly detectors.

## 2 RELATED WORK AND MOTIVATION

In this section, we will give a brief introduction to the existing AAD work and analyze their limitations from two aspects: (1) sample selection and (2) feedback incorporation.

### 2.1 SAMPLE SELECTION

Most AAD approaches (Siddiqui et al., 2018; Das et al., 2017; Zha et al., 2020; Li et al., 2019; Das et al., 2016) adopt the top-selection strategy. The anomalous samples, that are not ranked on the top initially by the base detector, would have little chance to be selected for feedback, and therefore can hardly be recalled subsequently. We show a real example using KDD-99 SA[1], which is a famous intrusion detection dataset. The dataset contains one normal class (96.7%) and 11 anomalous classes (3.3%) of various intrusion types. We applied the Isolation Forest (Liu et al., 2012) detector (a widely accepted one) to this dataset and found that the recall was around 0.28. We show the anomaly score distribution for the normal samples and three major intrusion types, respectively, in Fig. 2. Only the samples of two intrusion types, i.e., "neptune" and "satan", are assigned high anomaly scores ($0.60 \sim 0.70$). However, the samples of another major intrusion type "smurf" (accounts for 71.27% of all anomalous samples) are assigned relatively low anomaly scores ($0.50 \sim 0.55$), which is even below the anomaly scores of many normal samples (4168 normal samples vs. 15 "smurf" anomalies were assigned anomaly scores over 0.55). Under this circumstance, selecting the top samples only for feedback can hardly improve the recall for the "smurf" type. In LMADA, we consider both anomaly scores as well as the diversity of samples during the sample selection. In this way, samples

---

[1]https://archive.ics.uci.edu/ml/machine-learning-databases/kddcup99-mld/kddcup.data.gz

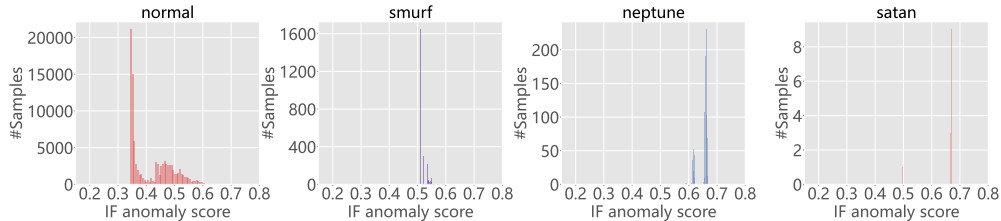

Figure 2: The anomaly score distribution of different classes in the KDD99-SA dataset.

not initially ranked on the top, like the "smurf" anomalies in our example, can have an opportunity to present to analysts.

## 2.2 Feedback Incorporation

How to incorporate feedback information is another focus of AAD. Das et al.(Das et al., 2017) added a set of adjustable weights to the random projections generated by LODA detector (Pevnỳ, 2016), by which the feedback can be incorporated. They also modified Isolation Forest (Liu et al., 2012) by adding weights to the tree paths, re-weighting the isolation score based on the feedback (Das et al., 2016). Siddiqui et al.(Siddiqui et al., 2018) extended the re-weighting strategy to the Generalized Linear Anomaly Detectors (GLAD) with the help of online convex optimization (Hazan et al., 2016). iRRCF-Active (Wang et al., 2020) also borrowed the above similar idea into iRRCF detector (Guha et al., 2016). In summary, the above methods require tailoring the weights specific to the certain model structure of different unsupervised detectors and then adjusting the weights with feedback-labeled samples by gradient descent. However, it is impractical for such a diverse range of unsupervised detectors as the modification is sophisticated and case-by-case. In LMADA, we propose a model-agnostic method to incorporate feedback information, regardless of the type of unsupervised detectors.

We also note that some AAD approaches have been proposed and attempted to support arbitrary base detectors. Meta-AAD (Zha et al., 2020) first extracts a set of transferable features based on $k$-neighbors to labeled instances and feeds them into a pre-trained meta-policy model for detection. GAOD (Li et al., 2019) leverages label spreading (Zhou et al., 2003), a graph-based semi-supervised model, to iteratively spread label information to neighbors. In summary, both AAD methods leverage neighborhoods of labeled instances to exploit feedback information but require persisting the entire dataset for neighboring sample retrieval. Therefore, the final tuned detection model would become increasingly heavier and heavier. In this paper, the feedback incorporation of LMADA is achieved by only a non-linear transformation, which is lightweight enough for real-world application.

## 3 Approach

In this section, we will elaborate on the details about LMADA. Following the general AAD workflow shown in Fig.1, LMADA consists of two components, i.e., sample selector and model tuner. In the sample selector, we consider the diversity in addition to the anomaly scores when recommending valuable samples for labeling. In the model tuner, we proposed a model-agnostic strategy to incorporate feedback information for arbitrary unsupervised detectors. It is achieved in a lightweight manner, only relying on a simple non-linear transformation.

## 3.1 Sample Selector

As discussed in Sec. 2.1, sample selection of AAD should consider the diversity of the selected samples in addition to the anomaly scores. The diversity here is not in terms of anomaly scores but in the distribution of the samples. In summary, our attempt is to select a subset of samples with high anomaly scores, and meanwhile, are dissimilar from each other. We use the example shown in Fig. 3 to illustrate this idea. There are two types of anomalies A and B that stray from the majority of samples. The anomaly scores (based on the Isolation Forest) are indicated by the colors. The

deeper the color, the higher the anomaly score. The selected samples are indicated by the blue cross markers. The number of selected samples is fixed as 20. Type-B anomalies are assigned relatively lower anomaly scores compared with type-A because they are more adjacent to the normal samples.

If we use the top-selection strategy, the selected samples would mostly come from type-A (as shown in the left subfigure of Fig.3), which may not cover the other types of anomalies. Therefore, the feedback would not help the AAD to recall more anomalies, e.g., type-B in this example. The desired sample selection is shown in the right subfigure of Fig.3, where the selector achieves a good coverage for all samples with relatively high anomaly scores. In this way, we can enhance the anomaly scores of all anomaly types, instead of only those originally ranked high by the base detector.

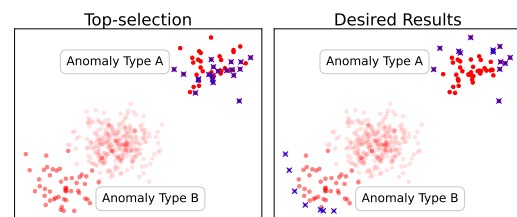

Figure 3: An illustration example of the top-selection and the desired sampling results.

Inspired by (Chen et al., 2018), we leverage a widely-adopted diversity sampling method, i.e., Determinantal Point Processes (DPP) (Kulesza et al., 2012), to achieve the above sampling target. We first introduce DPP in Sec. 3.1.1, and then describe how we balance the dual objectives, i.e., anomaly score and diversity, in Sec. 3.1.2.

### 3.1.1 DETERMINANTAL POINT PROCESSES (DPP)

The Determinantal Point Process (DPP) was originally introduced from fermion systems in thermal equilibrium (Macchi, 1975; Chen et al., 2018). Recently, it has been successfully applied to various machine learning tasks, e.g., image search (Kulesza & Taskar, 2011a), document summarization (Kulesza & Taskar, 2011b) and recommendation systems (Gillenwater et al., 2014). Given a dataset $\mathcal{D} = \{s_1, s_2, ..., s_n\}$, DPP aims to select a subset $\mathcal{C}$ from $\mathcal{D}$. Specifically, DPP constructs a real positive semidefinite (PSD) kernel matrix $\boldsymbol{L} \in \mathbb{R}^{n \times n}$ derived from $\mathcal{D}$. For each subset $\mathcal{C} \subseteq \mathcal{D}$, the probability of selecting $\mathcal{C}$ from $\mathcal{D}$, denoted as $P(\mathcal{C})$, is proportional to $\det(\boldsymbol{L}_\mathcal{C})$, where $\det(\boldsymbol{L}_\mathcal{C})$ is the determinantal value of the principal minor $\boldsymbol{L}_\mathcal{C}$. The objective of DPP is to derive $\mathcal{C}^*$ which maximizes the value of $\det(\boldsymbol{L}_\mathcal{C})$, shown in Eq.1. As an example, to achieve maximum diversity, the kernel matrix could be constructed as the pairwise similarity matrix (Kulesza et al., 2012).

$$\mathcal{C}^* = \mathrm{argmax}_{\mathcal{C} \subseteq \mathcal{D}} \det(\boldsymbol{L}_\mathcal{C}) \tag{1}$$

How to approximately solve this NP-hard problem (Ko et al., 1995) has been well studied in (Gillenwater et al., 2012; Han et al., 2017; Li et al., 2016; Chen et al., 2018) and we adopt the greedy algorithm proposed in (Chen et al., 2018) in our paper. We will introduce how to construct a specially tailored kernel matrix $\boldsymbol{L}$ for AAD in the next section.

### 3.1.2 KERNEL MATRIX CONSTRUCTION

In LMADA, we construct a kernel matrix $\boldsymbol{L}$, whose entries can be formally written as Eq.2,

$$\boldsymbol{L}_{ij} = \langle a_i r_i \boldsymbol{s}_i, a_j r_j \boldsymbol{s}_j \rangle = a_i a_j r_i r_j \langle \boldsymbol{s}_i, \boldsymbol{s}_j \rangle \tag{2}$$

where $a_i$ denotes the anomaly score uniformly re-scaled in the range of $[0, 1]$. It is used to motivate DPP to select samples with high anomaly scores. Meanwhile, we need to select diverse samples within and across feedback iterations. In each feedback iteration, the inner product $\langle \boldsymbol{s}_i, \boldsymbol{s}_j \rangle$ measures the pairwise similarity of all candidate samples, based on which DPP prefers dissimilar samples (Kulesza et al., 2012). As there are multiple feedback iterations, we expect the samples selected in the current iteration are also different from those sampled in previous iterations. To achieve so, we maintain a data pool $\mathcal{P}$ preserving the selected samples from the previous feedback iterations. The minimum distance between a candidate sample $\boldsymbol{s}_i$ and the selected samples cached in $\mathcal{P}$, is defined as the *feedback repulsion score* $r_i$, as shown in Eq.3.

$$r_i = \min(\{1 - \langle \boldsymbol{s}_i, \boldsymbol{s}_k \rangle \, | \forall \boldsymbol{s}_k \in \mathcal{P}\}) \tag{3}$$

From Eq.2, we can conclude that $\det(\boldsymbol{L}_\mathcal{C})$ is proportional to $a_i a_j r_i r_j$ and is inversely proportional to $\langle \boldsymbol{s}_i, \boldsymbol{s}_j \rangle$ among the selected samples in $\mathcal{C}$. In this way, it induces DPP to select more anomalous

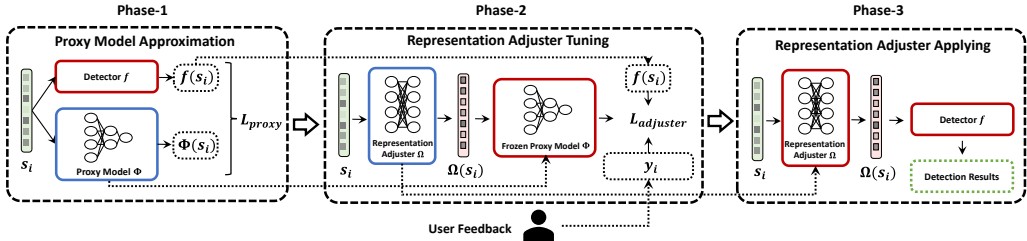

Figure 4: The overview of the model tuner. The blue boxes and red boxes denote the trainable/frozen components, respectively.

(i.e., higher $a_i a_j$) data points that are not adjacent to the previously selected examples (i.e., higher $r_i r_j$). Meanwhile, the data points are also distinguish enough from each other (i.e., lower $\langle s_i, s_j \rangle$). The qualitative analysis can be referred to Appendix Sec.A.1.

Theoretically, the complexity of constructing $L$ is $O(n^2)$, which is expensive for a large dataset. However, anomalous samples generally account for a small percentage of the whole dataset compared with the normal class (Zhao et al., 2019; Boukerche et al., 2020). For the instance in KDD99-SA dataset introduced in Sec.2.1, only 3.3% of samples belong to anomalies. It is unnecessary to regard all samples as candidates for the sample selector. Consequently, we construct the kernel matrix with only the pre-truncated top $\alpha\%$ samples ranked by their anomaly scores. In general, if $\alpha$ is small enough (e.g., $< 3\%$), the selected samples would be those with the highest anomaly scores, i.e., similar to the top-selection. On the other hand, if $\alpha$ is large (e.g., $> 30\%$), the selected samples would become too diverse to retrieve samples worthwhile for feedback. We will evaluate different $\alpha$ settings in Appendix Sec.A.8.

## 3.2 MODEL TUNER

After labeling the examples recommended by the sample selector, the model tuner focuses on how to incorporate newly labeled data points. The model tuner should be agnostic to the base unsupervised detectors. In other words, any unsupervised detection model can be easily integrated into our framework. To achieve this goal, we propose a three-phases model tuner in LMADA, as shown in Fig. 4. Firstly, we set up a neural network as the proxy model (Coleman et al., 2019) to mimic the behaviors of diverse base detectors. After that, a representation adjuster is added in front of the frozen proxy model to get trained based on the labeled samples. Finally, the tuned representation adjuster is used to transform the original samples into new representation vectors, which will be fed back to the base detector for re-scoring. The feedback continues for multiple iterations until the sampling budget is exhausted. The tuned representation adjuster can be applied as illustrated in the Phase-3 of Fig.4. Given a testing sample $s_i$, we first transform it into a new representation vector $h_i$ via the representation adjuster $\Omega(s_i)$. Then we directly feed $h_i$ into the base anomaly detector $f$ and get the final detection results $f(h_i)$. In this way, LMADA achieves feedback incorporation in a lightweight manner, only with a non-linear representation transformation.

### 3.2.1 PROXY MODEL APPROXIMATION

As introduced in Sec. 2.2, unsupervised detectors of various types pose a great challenge to model-agnostic AAD. There are significant differences between the model structures of different unsupervised detectors. Most existing AAD work (Siddiqui et al., 2018; Das et al., 2017; 2016; Wang et al., 2020) needs to specifically modify the internal structure of unsupervised detectors.

To tackle this problem, we utilize a deep neural network as the proxy model to approximate the behaviors of diverse unsupervised detectors. In this way, we can turn unsupervised detectors into gradient-optimizable neural networks, which facilitate the subsequent representation adjuster tuning (more details presented in Sec.3.2.2). As shown in Phase-1 of Fig. 4, we use the normalized anomaly scores $f(s_i)$ generated by the base detector as the pseudo-labels and set up a neural network $\Phi$ in parallel to fit them. The proxy model is composed of one input layer and multiple hidden layers followed by an output layer activated by the sigmoid function. The Mean-

Squared-Error (MSE) is adopted as the loss function during proxy model training, as shown in $\mathcal{L}_{proxy} = \sum_{i=1}^{b} \left( \Phi\left( \boldsymbol{s_i} \right) - f\left( \boldsymbol{s_i} \right) \right)^2$, where $b$ denotes the batch size.

After the proxy model training, the anomalous patterns that are captured by the base detectors have been learned by the proxy model, i.e., the proxy anomaly scores $\Phi\left( \boldsymbol{s_i} \right) \approx f\left( \boldsymbol{s_i} \right)$. The key point here is that the internal structures of different unsupervised detectors do not need to be considered in this training process.

### 3.2.2 REPRESENTATION ADJUSTER TUNING

In Phase-2, we devise a representation adjuster $\Omega$ in front of the proxy model to incorporate the feedback information. The representation adjuster is a simple non-linear transformation layer, which takes the original sample vector $\boldsymbol{s_i}$ as the input and transforms it into a new feature space but with the same dimensions, i.e., $\boldsymbol{h_i} = \Omega\left( \boldsymbol{s_i} \right) = \text{sigmoid}\left( \boldsymbol{W s_i} \right)$, where $\boldsymbol{h_i} \in \mathbb{R}^d$ and $\boldsymbol{s_i} \in \mathbb{R}^d$.

As shown in the middle of Fig.4, the transformed $\boldsymbol{h_i}$ will be fed into the trained proxy model $\Phi$ and generate the proxy anomaly score $\Phi\left( \boldsymbol{h_i} \right)$. Based on that, $\boldsymbol{W}$ will be updated under the loss function in Eq.4. The representation adjuster can be trained by a gradient descent optimizer because the subsequent proxy model (as shown in Fig. 4) is also a neural network. The parameters of the proxy model are frozen during the representation adjuster tuning phase.

$$\mathcal{L}_{adjuster} = \mathcal{L}_{feedback} + \mathcal{L}_{consolidation} + \eta \tag{4}$$

$\mathcal{L}_{adjuster}$ is composed of three components, i.e., feedback loss, consolidation loss and a regularization item $\eta$. $\mathcal{L}_{feedback}$ is used to fit the labeled samples in the data pool $\mathcal{P}$, as shown in Eq.5, where $y_i$ represents the feedback label (+1 for the anomalous class and -1 for the normal class) for the sample $\boldsymbol{s_i}$.

$$\mathcal{L}_{feedback} = -\sum_{i=1}^{b} y_i * \log\left( \Phi\left( \boldsymbol{h_i} \right) \right), \forall \boldsymbol{s_i} \in \mathcal{P} \tag{5}$$

Training with only a few labeled samples would make the representation adjuster biased toward the feedback labels but ignore the patterns already learned from the base detector. So we design another component inspired by (Li & Hoiem, 2017), i.e., $\mathcal{L}_{consolidation}$, that serves for consolidating the knowledge of the base unsupervised detector, as shown in Eq.6. $\tilde{\boldsymbol{h_i}}$ denotes the transformed sample representation in the last feedback iteration ($\tilde{\boldsymbol{h_i}} = \boldsymbol{s_i}$ in the first feedback iteration). It forces the proxy anomaly scores $\Phi\left( \boldsymbol{h_i} \right)$ of the remaining unlabeled samples to be stabilized around the original anomlay scores $f\left( \tilde{\boldsymbol{h_i}} \right)$ in the newly transformed feature space. We note that $\mathcal{L}_{consolidation}$ is not conducive to fitting $\mathcal{L}_{feedback}$ as the former tends to remain the original representation. To achieve a trade-off between them, we assign a weight for the consolidation loss of each sample. Intuitively, if an unlabeled sample $s_i$ is similar to the labeled samples in the feedback data pool $\mathcal{P}$, its consolidation loss should have a lower weight, reducing the constraints for fitting $\mathcal{L}_{feedback}$. On the contrary, those unlabeled samples, which are unlike the data points in $\mathcal{P}$, should be assigned a higher weight to enhance the influence of the consolidation loss. This intuition is fully aligned with the feedback repulsion score $r_i$ introduced in Sec.3.1.2 and we thus use it as the weight of consolidation loss.

$$\mathcal{L}_{consolidation} = \sum_{i=1}^{b} r_i * \left( \Phi\left( \boldsymbol{h_i} \right) - f\left( \tilde{\boldsymbol{h_i}} \right) \right)^2, \forall \boldsymbol{s_i} \notin \mathcal{P} \tag{6}$$

The last component is the penalty for feature space transformation because the extremely dramatic change to the original sample vectors is undesired. To achieve so, we set $\eta$ as $\sum_{i=1}^{b} ||\boldsymbol{h_i} - \boldsymbol{s_i}||^2$. More training details for the representation adjuster can be found in Appendix Sec.A.2.

## 4 EXPERIMENT

### 4.1 DATASETS AND SETTINGS

We evaluated our proposed method on 8 public datasets, including PageBlocks, Annthyroid, Cardio, Cover, KDD99-Http, Mammography, KDD99-SA, Shuttle, which are widely used by existing AAD

approaches (Siddiqui et al., 2018; Zha et al., 2020; Li et al., 2019; Das et al., 2017; 2019). The details of these datasets can be found in Appendix Sec. A.3. We run 5 feedback iterations and query 20 samples in each iteration. Same as the existing work, we used simulation instead of real user feedback since all the ground truth is known for these public datasets. The experimental environment and the parameters setting can be found in Appendix Sec. A.4 and Sec. A.5, respectively.

## 4.2 COMPARISON METHODS AND METRICS

We compared LMADA with three state-of-the-art AAD methods, i.e., FIF (Siddiqui et al., 2018), Meta-AAD (Zha et al., 2020), and GAOD (Li et al., 2019). FIF adds a set of weights to the tree branches of the Isolation Forest detector and tunes them via online convex optimization with feedback information. GAOD utilizes the semi-supervised method (label spreading Zhou et al. (2003)) to consume user feedback. Both of the above approaches adopt the top-selection strategy. Meta-AAD extracts a set of transferable features for a pre-trained meta-policy detection model, considering both long-term and short-term benefits in querying feedback.

We use F1-Score Curve to evaluate the effectiveness of different AAD methods. Specifically, we calculate F1-Score on the entire dataset after finishing an iteration of feedback. Besides, we also calculate the Area-Under-Curve (AUC) (Ling et al., 2003) of the F1-Score Curve.

## 4.3 COMPARISON EXPERIMENT RESULTS

We compared our proposed method with three state-of-the-art AAD approaches and the results are illustrated in Fig. 5. For fairness, we used Isolation Forest as the base detector because it was adopted by all the comparison methods (Zha et al., 2020; Siddiqui et al., 2018; Li et al., 2019). To ensure reproducibility, we repeated our experiments 10 times on each dataset and plotted the average F1-Score and the standard error bar (Altman & Bland, 2005). The AUC value of each F1-Score Curve is shown in the legend.

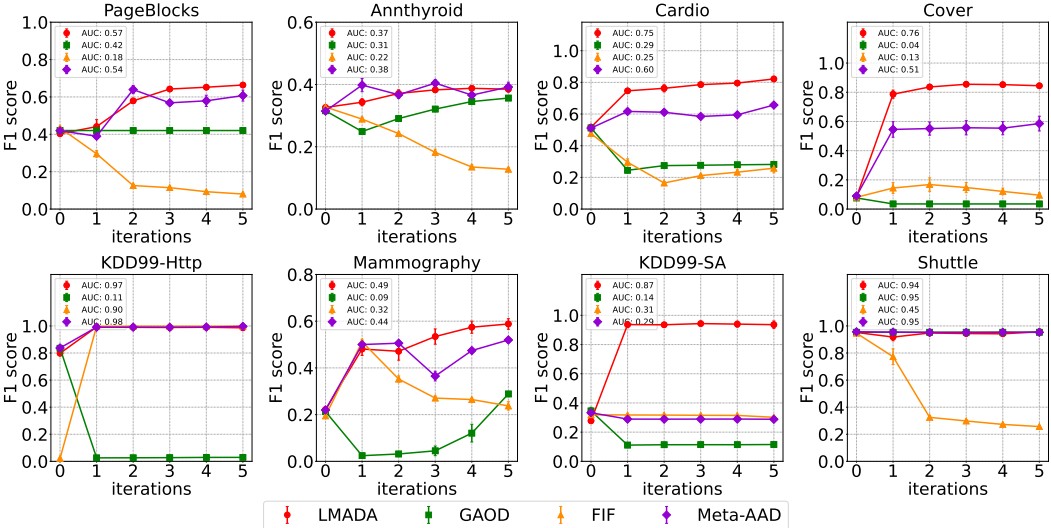

Figure 5: The experiment results comparing LMADA with the state-of-the-art AAD methods.

From the results, we can confirm that LMADA performs better than other AAD methods. With 20 feedback samples per iteration, LMADA achieved consistently higher F1-Score on most datasets. Especially on KDD99-SA, Cover, and Cardio datasets, LMADA boosted the F1-Score of the base detector by an average of 144% to 0.80+ after 5 feedback iterations. For PageBlocks, Annthyroid, and Mammography datasets, LMADA also increased the F1-Score by 60% on average, significantly outperforming other AAD models. As for the KDD99-Http and Shuttle dataset, we can see that the initial performance of the base detector has reached a relatively high level. Under this circumstance, LMADA also can hold a high detection accuracy, exhibiting its robustness.

Among the comparison methods, Meta-AAD performed much better than the other two because it utilizes reinforcement learning to learn a meta-policy for feedback querying, rather than simply picking up the samples with the highest anomaly scores. However, the diversity of samples is not taken into account explicitly, resulting in relatively lower performance compared with LMADA (e.g. 0.29 AUC of Meta-AAD vs. 0.87 AUC of LMADA in KDD99-SA dataset). FIF and GAOD even had difficulty preserving the upward trend of their F1-Score curves, although more feedback samples were added. As we discussed in Sec.2.1, the top-selection strategy of both methods hinders the exploration of the lower-ranked anomalous samples. Moreover, their detectors were tuned to over-fit the scarce feedback-labeled samples, leading to a decreasing recall. We have verified this in Appendix Sec. A.9.

## 4.4 MODEL-AGNOSTIC EVALUATION

We target to propose a model-agnostic AAD approach, which can be easily extended to arbitrary unsupervised detectors. As such, we evaluated the effectiveness of LMADA under five different but commonly-used unsupervised detectors, including AutoEncoder (Vincent et al., 2010), PCA (Shyu et al., 2003), OCSVM (Schölkopf et al., 2001), LODA (Pevnỳ, 2016; Das et al., 2016), and IF. The experimental settings are the same as that in Sec.4.3 and the results are shown in Fig. 6.

From these figures, we can conclude that LMADA works well on different unsupervised detectors. It can consistently improve the F1-Score on all eight datasets whatever the base detector is adopted. More than that, we also found that the performance gains achieved by LMADA vary with different unsupervised detectors. Taking the KDD99-Http dataset as an example, we can see that LODA performs much worse than the other base detectors at the beginning (F1-Score 0.02 compared to ~0.82 of the other detectors). Even so, LMADA was also able to improve the performance of LODA from 0.02 to 0.96 after 5 iterations. We also noted that the variance of its results is significantly larger than the others. The reason is that LODA is inaccurate and unstable on KDD99-Http dataset, making it difficult to provide effective information for the sample selector and the model tuner. These experiment results confirm that the initial performance of base detectors has a great influence to AAD approaches.

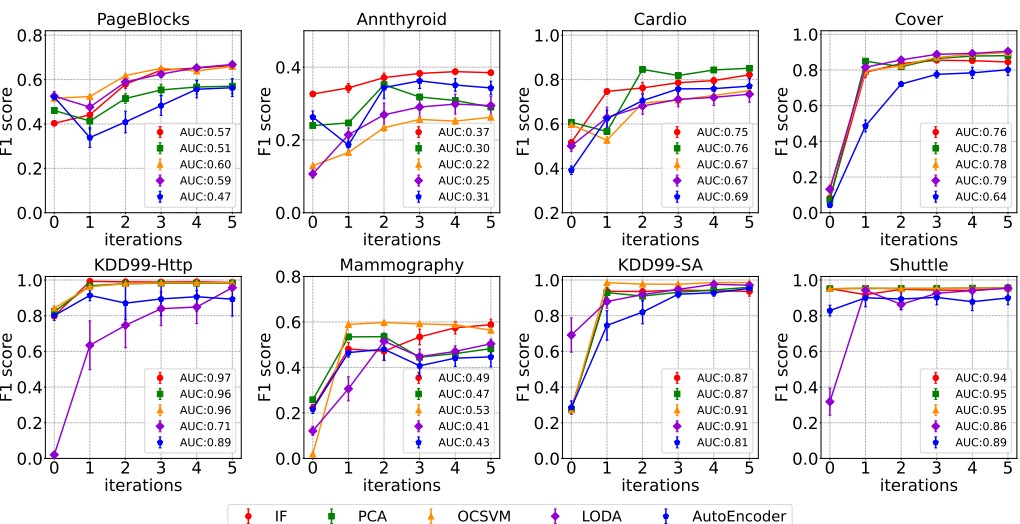

Figure 6: The results of LMADA under different base unsupervised detectors.

## 4.5 SAMPLE SELECTOR VALIDATION

In this section, we validated the effectiveness of our proposed sample selector in LMADA. As we discussed in Sec. 2.1, diversity plays a critical role in AAD. In order to verify this point, we conducted an ablation study on the KDD99-SA dataset. In this dataset, 11 anomalous classes and the normal class are well annotated separately so that we can study how samples would be selected by different sampling strategies. We compared our proposed sampling method with the commonly-used

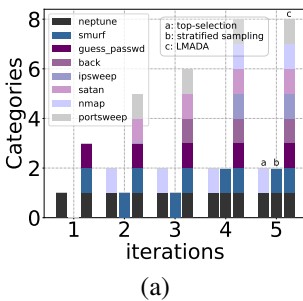 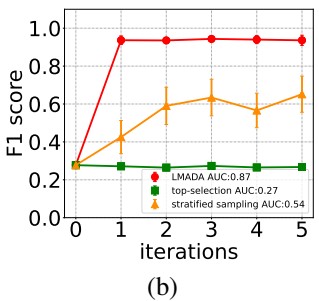 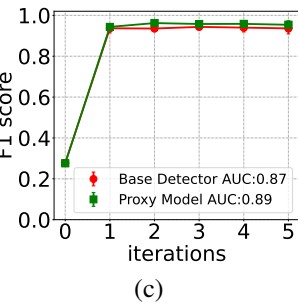

|            |            |            |
|    (a)     |    (b)     |    (c)     |

Figure 7: (a) The selected anomaly classes under different sampling strategies. (b) The F1-Score comparison results under different sampling strategies. (c) The F1-Score comparison results between the proxy model and the base detector.

top-selection strategy (Das et al., 2017; 2016; Siddiqui et al., 2018), and the stratified sampling described in (Guha et al., 2016) (i.e., divide samples into $g$ groups based on their anomaly scores and then select examples randomly from each group). The model tuner is fixed. The selected anomalous classes under these settings and their corresponding improved F1-Scores are shown in Fig. 7(a) and Fig. 7(b), respectively.

From Fig.7(a), we can see that the sample selector of LMADA is able to cover more anomaly classes, compared with the other two sampling strategies. Furthermore, we also confirm the necessity of the diversity-aware selection from Fig.7(b) since our sample selector achieved much higher F1-Scores than those under the top-selection or the stratified sampling methods. For example, in the first feedback iteration, our proposed sample selector chose "smurf" samples (shown in blue color) for feedback, which were missed by the other two. As we stated in Sec.2.1, "smurf" samples were not assigned high anomaly scores by the base detector (IF) but they actually account for 71.27% of all anomalies. Therefore, we can see that F1-Score can be significantly improved from 0.28 to 0.94 with labeled "smurf" anomalies, while the other two strategies failed to achieve this high F1-Score. The complete results on all datasets can be found in Appendix Sec. A.6.

## 4.6 MODEL TUNER VALIDATION

In this section, we will present the effectiveness of our proposed model tuner. As introduced in Sec.3.2, the transformed representations $h_i$ are trained based on the proxy model but will be fed back to the base unsupervised detector to get the final anomaly scores. We aim to study how large the difference between the anomaly scores generated by the base detector $f(h_i)$ and the proxy model $\Phi(h_i)$, respectively. We also conducted this ablation experiment on the KDD99-SA dataset and the results are exhibited in Fig. 7(c).

This figure shows that there is only a narrow gap in F1-Scores between the proxy model (green line) and the base unsupervised detector (red line). It manifests that the proxy model has captured the knowledge learned by the base detection method as they produced similar anomaly scores. As such, the transformed representations $h_i$ trained via the proxy model can be smoothly transferred to the base unsupervised detector. The complete experimental results on all datasets can be referred to Appendix Sec. A.7.

## 5 CONCLUSION

In this paper, we propose LMADA, a lightweight, model-agnostic and diversity-aware active anomaly detection method. In the sample selector of LMADA, we take the anomaly scores as well as the diversity of samples into account, unlike most existing AAD work that solely picks the most anomalous ones for feedback querying. In the model tuner of LMADA, we propose a model-agnostic strategy to incorporate feedback information, regardless of the type of unsupervised detector. It can be achieved by a lightweight non-linear transformation. Through the extensive evaluation on 8 public AD datasets, we show that LMADA can achieve 74% F1-Score improvement on average, significantly outperforming other comparative AAD approaches.

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

# A  APPENDIX

## A.1  THE QUALITATIVE ANALYSIS OF EXTENDED DPP IN SAMPLE SELECTOR

The kernel matrix $\boldsymbol{L}$ is shown as Eq.2. As introduced in Sec.3.1.1, we aim to select a subset $\mathcal{C}$ with highest $\det(\boldsymbol{L}_{\mathcal{C}})$. The principal minor $\boldsymbol{L}_{\mathcal{C}}$ is as follows.

$$
\begin{bmatrix}
a_1^2 r_1^2 \langle \boldsymbol{s}_1, \boldsymbol{s}_1 \rangle & \cdots & a_1 a_j r_1 r_j \langle \boldsymbol{s}_1, \boldsymbol{s}_j \rangle & \cdots & a_1 a_{|\mathcal{C}|} r_1 r_{|\mathcal{C}|} \langle \boldsymbol{s}_1, \boldsymbol{s}_{|\mathcal{C}|} \rangle \\
a_2 a_1 r_2 r_1 \langle \boldsymbol{s}_2, \boldsymbol{s}_1 \rangle & \cdots & a_2 a_j r_2 r_j \langle \boldsymbol{s}_2, \boldsymbol{s}_j \rangle & \cdots & a_2 a_{|\mathcal{C}|} r_2 r_{|\mathcal{C}|} \langle \boldsymbol{s}_2, \boldsymbol{s}_{|\mathcal{C}|} \rangle \\
\vdots & \ddots & \vdots & \ddots & \vdots \\
a_i a_1 r_i r_1 \langle \boldsymbol{s}_i, \boldsymbol{s}_1 \rangle & \cdots & a_i a_j r_i r_j \langle \boldsymbol{s}_i, \boldsymbol{s}_j \rangle & \cdots & a_i a_{|\mathcal{C}|} r_i r_{|\mathcal{C}|} \langle \boldsymbol{s}_i, \boldsymbol{s}_{|\mathcal{C}|} \rangle \\
\vdots & \ddots & \vdots & \ddots & \vdots \\
a_{|\mathcal{C}|} a_1 r_{|\mathcal{C}|} r_1 \langle \boldsymbol{s}_c, \boldsymbol{s}_1 \rangle & \cdots & a_{|\mathcal{C}|} a_j r_{|\mathcal{C}|} r_j \langle \boldsymbol{s}_{|\mathcal{C}|}, \boldsymbol{s}_j \rangle & \cdots & a_{|\mathcal{C}|}^2 r_{|\mathcal{C}|}^2 \langle \boldsymbol{s}_{|\mathcal{C}|}, \boldsymbol{s}_{|\mathcal{C}|} \rangle
\end{bmatrix}
\tag{7}
$$

The $\det(\boldsymbol{L}_{\mathcal{C}})$ can be calculated in Eq 8.

$$
\det(\boldsymbol{L}_{\mathcal{C}}) = \sum (-1)^{\tau(p_1, p_2, \dots p_{|\mathcal{C}|})} L_{1 p_1} L_{2 p_2} \cdots L_{|\mathcal{C}| p_{|\mathcal{C}|}}
\tag{8}
$$

where $p_1, p_2, \dots p_{|\mathcal{C}|}$ denote all permutations of $\{1, 2, \dots |\mathcal{C}|\}$, and $\tau(p_1, p_2, \dots p_{|\mathcal{C}|})$ represents the reverse order number of $p_1, p_2, \dots p_{|\mathcal{C}|}$. According to Eq.2, $\det(\boldsymbol{L}_{\mathcal{C}})$ can be further expanded as Eq. 9

$$
\det(\boldsymbol{L}_{\mathcal{C}}) = \prod_{i=1}^{|\mathcal{C}|} a_i^2 r_i^2 \sum (-1)^{\tau(p_1, p_2, \dots p_{|\mathcal{C}|})} \langle \boldsymbol{s}_1, \ \boldsymbol{s}_{p_1} \rangle \langle \boldsymbol{s}_2, \ \boldsymbol{s}_{p_2} \rangle \cdots \langle \boldsymbol{s}_{|\mathcal{C}|}, \ \boldsymbol{s}_{p_{|\mathcal{C}|}} \rangle
\tag{9}
$$

$$
= \prod_{i=1}^{|\mathcal{C}|} a_i^2 r_i^2 \cdot \left| \det \left( \left[ \boldsymbol{s}_1, \boldsymbol{s}_2, \dots \boldsymbol{s}_{|\mathcal{C}|} \right]^\top \left[ \boldsymbol{s}_1, \boldsymbol{s}_2, \dots \boldsymbol{s}_{|\mathcal{C}|} \right] \right) \right|
\tag{10}
$$

$$
= \prod_{i=1}^{|\mathcal{C}|} a_i^2 r_i^2 \cdot \left( \boldsymbol{s}_1 \otimes \boldsymbol{s}_2 \otimes \dots \otimes \boldsymbol{s}_{|\mathcal{C}|} \right)^2 = \prod_{i=1}^{|\mathcal{C}|} a_i^2 r_i^2 \cdot V^2
\tag{11}
$$

$\prod_{i=1}^{|\mathcal{C}|} a_i^2 r_i^2$ is the common factor extracted from $\det(\boldsymbol{L}_{\mathcal{C}})$. As such, we can conclude that $\det(\boldsymbol{L}_{\mathcal{C}})$ is proportional to $a_i$ and $r_i$, inducing DPP to select samples that have high anomaly scores and are different from those have already been selected in the data pool $\mathcal{P}$.

The second term, $\sum (-1)^{\tau(p_1, p_2, \dots p_{|\mathcal{C}|})} \langle \boldsymbol{s}_1, \ \boldsymbol{s}_{p_1} \rangle \langle \boldsymbol{s}_2, \ \boldsymbol{s}_{p_2} \rangle \cdots \langle \boldsymbol{s}_{|\mathcal{C}|}, \ \boldsymbol{s}_{p_{|\mathcal{C}|}} \rangle$, can be further rewrote as the exterior product form $\left( \boldsymbol{s}_1 \otimes \boldsymbol{s}_2 \otimes \dots \otimes \boldsymbol{s}_{|\mathcal{C}|} \right)^2$ shown in Eq.11. According to the definition of exterior product (Browne, 2012), it geometrically represents the volume $V$ of the parallel polyhedron spanned by vectors $\{ \boldsymbol{s}_1, \boldsymbol{s}_2, \dots \boldsymbol{s}_{|\mathcal{C}|} \}$. Consequently, the more dissimilar they are, the larger the volume $V$ of the spanned polyhedron is, the larger $\det(\boldsymbol{L}_{\mathcal{C}})$ is.

## A.2  LABELED SAMPLES OVERSAMPLING

In the model tuner, we use the labeled samples to train the representation adjuster. Nevertheless, compared to the unlabeled samples, the feedback-labeled samples only account for a tiny percentage of the overall dataset (e.g., 20 samples per iteration vs. 286048 samples in total of the Cover dataset). Therefore, we need to over-sample the labeled samples in each training batch to improve the utilization of such a few feedback samples, so that we can fully exploit the feedback information and accelerate the loss convergence. Half of each training batch are labeled samples, which are repeatedly drawn from the data pool $\mathcal{P}$, and the other half are unlabeled samples, which are randomly sampled from the all unlabeled samples.

## A.3  DATASETS INFORMATION

We used eight public datasets for the evaluation. PageBlocks, Annthyroid, Cardio, Cover, Mammography, Shuttle are available in ODDS [2]. KDD99-Http and KDD99-SA are available in UCI Machine

---

[2] http://odds.cs.stonybrook.edu/

Table 1: The Detailed Information of Experimental Datasets

| Datasets | Samples | Dimension | Anomaly Number | Anomaly Rate |
|---|---|---|---|---|
| PageBlocks | 5393 | 10 | 510 | 9.46% |
| KDD99-SA | 100655 | 95 | 3377 | 3.36% |
| Annthyroid | 7200 | 6 | 534 | 7.42% |
| Cardio | 1831 | 21 | 176 | 9.61% |
| Cover | 286048 | 10 | 2747 | 0.96% |
| KDD99-Http | 58725 | 3 | 2209 | 3.76% |
| Mammography | 11183 | 6 | 260 | 2.32% |
| Shuttle | 49097 | 9 | 3511 | 7.15% |

Learning Repository[3]. PageBlocks can be referred to ADBench [4]. The detailed information of these datasets is shown in Table.1. The number of samples ranges from 1.8K to 286K and the anomaly rate is spanning from 0.96% to 9.61%.

### A.4 EXPERIMENT ENVIRONMENT

We built LMADA based on PyTorch 1.12.0 (Paszke et al., 2019) and used base unsupervised anomaly detectors implemented in PyOD 1.0.3 (Zhao et al., 2019). In our experiments, we set up a Virtual Machine (VM) with 64 Intel(R) Xeon(R) Platinum 8370C CPU @ 2.80GHz processors and 256GB RAM. The operating system is Ubuntu-20.04. In the VM, we had an NVIDIA Tesla M40 GPU with CUDA 11.4 for deep learning model training.

### A.5 EXPERIMENT SETTING DETAILS

**LMADA:** For the sample selector of LMADA, we set the pre-truncation rate $\alpha = 10\%$. We introduce two hyper-parameters $\lambda$ and $\gamma$ to adjust the preference of anomaly score and diversity ($\boldsymbol{L}_{ij} = (a_i a_j)^\lambda (r_i r_j \langle \boldsymbol{s}_i, \boldsymbol{s}_j \rangle)^\gamma$). In the experiments, we set $\lambda = 1$ and $\gamma = 1$. In the model tuner, we utilized the Adam optimizer (Kingma & Ba, 2014) and set the epoch number to 10, the learning rate to 0.01, and the batch size to 512, for both the proxy model approximation phase and the representation adjuster tuning phase. The size of the proxy model hidden layer is set to 64. Specifically for SA dataset, we performed dimension reduction (Carreira-Perpinán, 1997) on it because it is characterized by its high feature dimensions and sparsity.

**Meta-AAD:** We used the source code available in the link provided by the original paper[5]. We utilized 12 datasets (including toy, yeast, glass, ionosphere, lympho, pima, thyroid, vertebral, vowels, wbc, wine, yeast) for meta-policy training in our experiment. All the datasets are available in the released code repository [6]. After that, we directly applied the trained meta-policy to the targeted 8 public datasets. We borrowed the the default settings from the original paper in our experiments: rollout steps $T = 128$, entropy coefficient $c_2 = 0.01$, learning rate $lr = 2.5 \times 10^{-4}$, value function coefficient $c_1 = 0.5$, $\lambda = 0.95$, clip range $\epsilon = 0.2$, balance parameter $\gamma = 0.6$.

**FIF:** We used the source code released in the link provided by the original paper [7]. We chose the Log-Likelihood loss function for FIF in the experiment. We set the type of regularizer $w = 2$ and the learning rate $a = 1$.

**GAOD:** We implemented GAOD according to Li et al. (2019) by ourselves because lacking the released source code. We set the number of nearest neighbors $k = 30$ and the learning rate of label spreading $\alpha = 0.995$. The standard deviation of Gaussian function $\sigma$ is set to half of the 95-percentile of k-th nearest neighbor distances.

---

[3]https://archive.ics.uci.edu/ml/machine-learning-databases/kddcup99-mld/kddcup.data.gz

[4]https://github.com/Minqi824/ADBench

[5]https://github.com/daochenzha/Meta-AAD

[6]https://github.com/daochenzha/Meta-AAD/tree/master/data

[7]https://github.com/siddiqmd/FeedbackIsolationForest

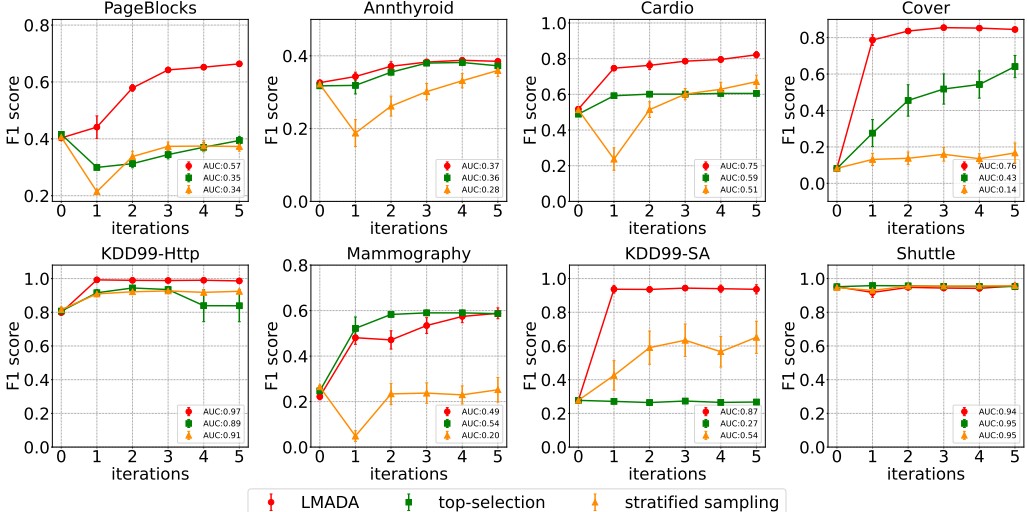

Figure 8: The complete results of sample selector validation.

We note that the pairwise distance matrix is required for Meta-AAD and GAOD (for neighborhood retrieval). As such, both approaches would fail to work under large data volume due to the high space complexity ($O(n^2)$). Taking the largest dataset Cover as an example (shown in Table.1), the pairwise distance matrix would consume 610 GB memory in theory, which would trigger the Out-Of-Memory (OOM) problem in our experiment environment. Therefore, we only keep the top 50% and 20% samples for KDD99-SA and Cover, respectively, based on the anomaly scores produced by the base detector. Only these samples are involved in the feedback incorporation of Meta-AAD and GAOD.

## A.6 The complete results of sample selector validation

We illustrated the sample selector validation results on all 8 datasets in Fig.8. Our sampling strategy outperforms other sampling methods on most datasets. Compared with the results of FIF and GAOD shown in Fig.5, we also found that our proposed method still achieved much better F1-Scores even using the top-selection strategy in the same manner. It confirms the effectiveness of our proposed model tuner on the other side.

## A.7 The complete results of model tuner validation

We show the model tuner validation results on all eight datasets in Fig.9. From these figures, we confirm the conclusion in Sec. 4.6. The proxy model has captured the knowledge learned by the base detection method as they produced similar anomaly scores. As such, the transformed representation $h_i$ can be directly fed into the base detector.

## A.8 Effectiveness of Pre-Truncation in Sample Selector

In Sec. 3.1.2, we introduced the pre-truncation to improve the sampling efficiency. In this section, we aim to validate its effectiveness in the sample selector. Specifically, we adjusted $\alpha$ from 1% to 60%. We recorded the running time and its corresponding AUC of F1-Score Curve under different $\alpha$ values, which are shown in Fig.10. From the left figure of Fig. 10, we can draw a conclusion that the running time can be significantly reduced by more pre-truncation. For example, the running time can be saved in half if we adjust $\alpha$ from 50% to ~6%. Moreover, from the right figure of Fig. 10, we can see that the AUC of F1-Score arises when $\alpha < 10\%$ and then gradually drops when we keep increasing $\alpha$. As we have discussed in Sec. 3.1.2, it manifests that either a too broad or a too narrow set of candidate samples leads to suboptimal feedback querying. Generally speaking, we set $\alpha$ around the estimated contamination ratio, such as 10%.

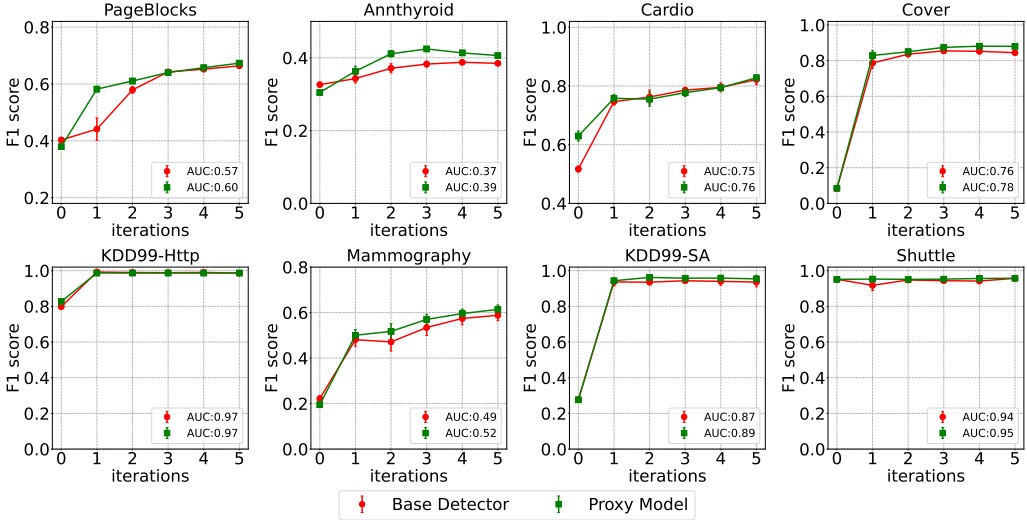

Figure 9: The complete results of model tuner validation.

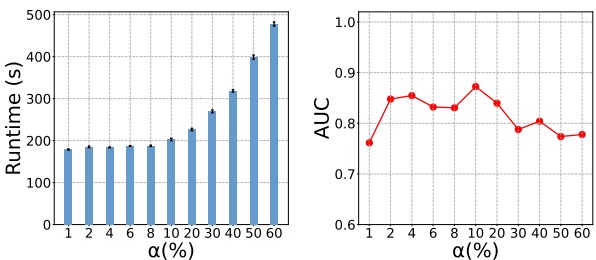

Figure 10: F1-Score Curve AUC and running time comparison under different $\alpha$.

## A.9 EXPLOARATION OF OVER-FITTING PROBLEM

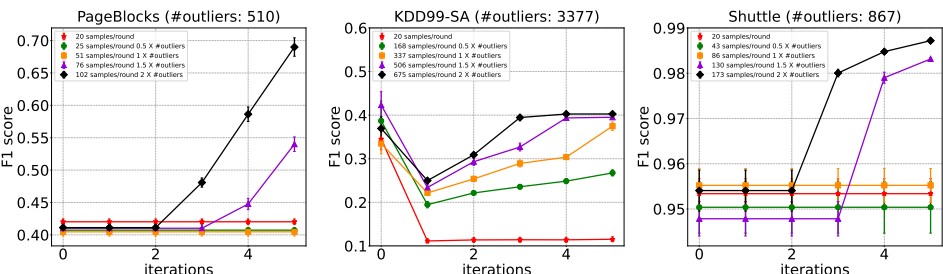

Figure 11: GAOD evaluation results under the increased number of queried samples.

In Sec.4.3, we found that the comparison methods performed much worse than LMADA. From the feedback incorporation perspective, it is caused by the overfitting to the few top-ranked samples (see Sec.1). To verify this point, we take GAOD as an example and gradually increase the number of querying samples in each feedback iteration to mitigate the overfitting problem. We rerun GAOD on three datasets (PageBlocks, Shuttle and KDD99-SA), where it did not perform well. According to the settings described in the original GAOD paper, the size of the data pool should be set to 2 × #outliers (Li et al., 2019). Therefore, we enlarged the data pool size spanning from 0.5 to 2 × #outliers by a stride of 0.5. From the results shown in Fig. 11, we see that GAOD can only achieve improvements in F1-Score with at least 0.5× #outliers (e.g., the number of queried samples reaches 168 per iteration in KDD99-SA dataset, which is far beyond our proposed approach with 20 per iteration). Therefore, it requires a significantly larger labeling effort.

## A.10 QUERY NUMBER EXPLORATION

We conducted the comparison experiment under different query numbers per feedback iteration (1, 5, 10, 20) on KDD-SA dataset, which can be found in Fig.12. From the figure, we can see that LMADA can achieve a consistent performance improvement, even with only 1 sample per iteration. On the contrary, the F1-Scores of FIF/GAOD/Meta-AAD fail to increase because they only select the top-ranked samples for updating the model, ignoring the low-ranked anomaly samples, such as the "smurf" type (as we presented in Sec.2.1).

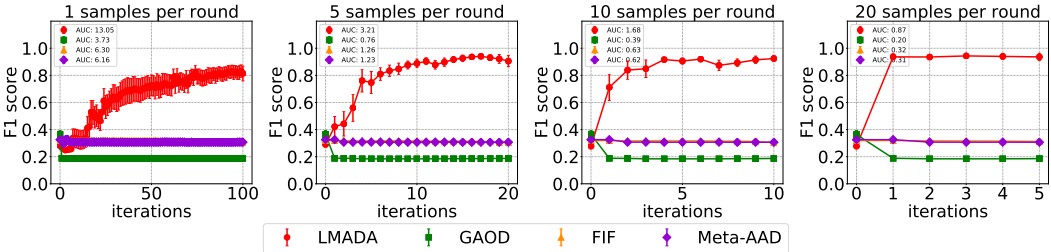

Figure 12: Comparison results under different query number per feedback iterations

## A.11 ADDITIONAL EXPERIMENT

We add the experimental results of the top-random query strategy in Fig.13, which represents a random selection from samples with high anomaly scores. From the results, we can conclude that our sampling method significantly outperforms the top-random on PageBlocks, Cardio, Cover, Mammography, KDD99-SA datasets and achieve similar performance on Annthyroid, KDD99-Http, and Shuttle datasets. Moreover, it is worth noting that the variance of the top-random strategy is much larger than that of ours.

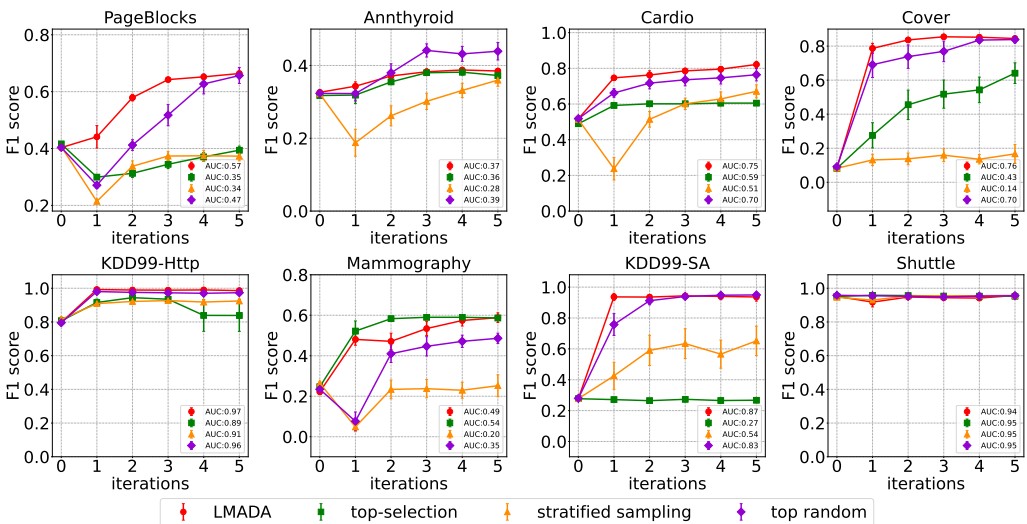

Figure 13: The F1-Score comparison results under different sampling strategies (including top random strategy).

