# OpenReview forum: "Towards Lightweight, Model-Agnostic and Diversity-Aware Active Anomaly Detection"
_ICLR.cc/2023/Conference — ICLR 2023 poster_

### Official Review · Reviewer_SZBd · 2022-10-23

**Confidence:** 4
**Correctness:** 3
**Technical Novelty And Significance:** 2
**Empirical Novelty And Significance:** 3
**Recommendation:** 5

**Clarity, Quality, Novelty And Reproducibility:**

The writing is clear. The idea of considering diversity samples is good but it simply uses the existing DPP to solve the problem. It is hard to reproduce the results because there is no source codes or network details provided in this paper.

**Strength And Weaknesses:**

Strength:
1.	It considers the diversity of samples instead of anomaly score top-selection strategy for the feedback querying, preventing the over-fitting situation to the top-ranked samples
2.	The proposed method shields the details of different unsupervised detectors and achieves lightweight feedback incorporation, only via a non-linear representation transformation.
3.	It conducts extensive experiments on eight public datasets and outperforms other comparative AAD approaches

Weaknesses:
1.	The proposed sampling method is somewhat incremental as it simply uses an existing method, DPP, to select samples. The authors may need to re-summarize the main contributions of the paper. The contributions are not clear.
2.	The proposed sampling method making the pre-truncated ratio alpha too important. It takes efforts to select such alpha technically by extensive experiments. And such alpha may differ in various dataset. For example, as shown in Appendix. A.8, when the ratio changed from 10% to 8%, the performance will decrease by 5%, which is too sensitive.
3.	It lacks the comparison of different sampling methods from selecting samples from the top alpha samples, like simple random sampling. I mean, the proposed method uses DPP to select diverse samples from top 10% anomaly data, but simply using random sampling to select samples from the same top 10% anomaly data can also guarantee the data diversity.
4.	The experiments of LAMDA under different base unsupervised detectors only consider classical unsupervised detectors like PCA, OCSVM. When using deep model, I am not sure whether the proxy model could mimic the detector or the cost in time and memory will be too large.
5.	This paper uses a proxy model to imitate different anomaly detectors. Authors claim that it is model agnostic. However, when using different anomaly detectors, the learned parameters in the proxy models are different. Thus, I think it is not model agnostic.

**Summary Of The Paper:**

This paper proposed a lightweight, model-agnostic and diversity-aware active
anomaly detection method. It takes the diversity of samples into account and designs a diversity-aware sample selector powered by determinantal point process. It proposes a model-agnostic strategy to incorporate feedback information, which approximates diverse unsupervised detectors with a unified proxy model.


**Summary Of The Review:**

As listed in the strength and weaknesses, the experiments are not convincing. It would be better if authors can add supplementary experiments on the model-agnostic evaluation and sample selector validation.

---

> ### Author Response · Authors · 2022-11-17
> **Response**
>
> We thank you for the valuable comments. Our response is as below.
>
> Q1: the contribution of the sample selector
>
> DPP is a framework for diversity sampling. For the feedback sampling phase, our contribution is to combine DPP with the anomaly score and repulsion score for a better trade-off between the diversity and anomaly scores. Without this, DPP is not appropriate for AAD tasks.
>
> Q2: question about the sensitivity of alpha
>
> We calculated the average value and the standard variance of AUC scores under different alpha. It is 0.833±0.035. So, the AUC scores stay quite stable with different alpha settings. As we said in Sec. A.5, alpha is uniformly set to 10% for all 8 datasets. The datasets have diverse anomaly ratio, but we achieved consistent promising results with respect to F1-Score, which double confirmed that the pre-truncated ratio is not very sensitive.
>
> Q3:  random selection on top-ranked samples
>
> We added experiment result of the query strategy you mentioned into Fig.13, which denoted as top-random. From the results, we can conclude that our sampling method significantly outperforms the top-random on PageBlocks/Cardio/Cover/Mammography/KDD99-SA datasets and achieve similar performance on Annthyroid/ KDD99-Http/Shuttle datasets. Moreover, it is worth noting that the variance of top-random strategy is much larger than that of ours.
>
> Q4: using deep learning detector as base AD
>
> We added new experimental results of LMADA based on Auto Encoder (AE) (https://pyod.readthedocs.io/en/latest/pyod.models.html#pyod.models.auto_encoder.AutoEncoder ), a deep learning base unsupervised detector, in Fig.14. From the figure, we can conclude that LMADA can also achieve significant performance improvement on all 8 datasets.
> As we stated in Sec.1 and Sec.2.2, our approach is model agnostic. Only the input samples and the output anomaly scores generated by the base AD are leveraged by LMADA, regardless of which kind of base AD is used. Therefore, the cost of LMADA will not be large under deep learning based AD. For example, we tested LMADA itself on KDD-SA dataset. It cost 127.16s (the average value after running 10 times experiments) when using AE and 123.19s under IF. Actually, the end-to-end training cost is more relevant to the base AD itself. It is also worth noting that the base AD is trained only ONCE and would not be re-trained during the model tuner updating phase.
>
> Q5: questions about “model agnostic”
>
> We have explicitly explained “model agnostic” in Sec.2.2. It means that we do not need to tailor the AAD method for different base AD case by case.  The learned parameters of the proxy model are not required to be identical. We use the same proxy model architecture for different datasets and set the hyper-parameters uniformly. The learned parameters are automatically obtained by training, without any prior knowledge of base AD.
>
> Q6: question about source code and network details
>
> We will release our source code and experimental datasets. The network architecture was introduced in Sec.3.2.1 and Sec.3.2.2 in detail. The hyper-parameter settings of the network were also elaborated in Appendix Sec.A.5.

---

> ### Comment · Area_Chair_YAMX · 2022-11-21
> **Any comments to the responses from authors?**
>
> Dear Reviewer SZBd,
>
> Thank you very much for your informative review.  The authors have provided responses to your concerns.  How did they change your evaluation, particularly on contributions and experimental support?

---

### Official Review · Reviewer_6kFK · 2022-10-24

**Confidence:** 3
**Correctness:** 4
**Technical Novelty And Significance:** 3
**Empirical Novelty And Significance:** 3
**Recommendation:** 8

**Clarity, Quality, Novelty And Reproducibility:**

The paper is organized and written well. The paper is novel in the sense that the authors identified two issues in AAD, and manage to solve them through an interesting framework that makes itself distinct from conventional solutions. The details are well elaborated, based on which the paper can be reproduced.

**Strength And Weaknesses:**

Strength:
* An interesting diversity-aware sample selector that not only considers the base detector scores but also the diversity of the samples was proposed and developed.
* A novel model-agnostic tuner is developed to integrate both the base model and user feedback.
* More specifically, first, a proxy model is developed to mimic the base detector. Second, the proxy network is frozen, and a new representation adjustor is added to learn new feature space, which seems a concise and effective design.

Weakness:
* There is no significant weakness identified in this paper. While the sample selector might be time-consuming and the time complexity of Determinantal Point Process (DPP) is O(n^2), the run time, in reality, will most likely not be overwhelming due to the limited amount of anomaly samples. Another minor to consider is adding additional experiments, although the current meta-ADD dataset has 24 tasks.


**Summary Of The Paper:**

The paper focuses on the two critical issues in ADD problem and proposes a lightweight, model-agnostic, and diversity-aware AAD method. The first issue is the negligence of lower-ranked samples in active learning, and the second issue is the generalization of the specified unsupervised detector to other models given different datasets or tasks. The new models manage to address both issues and demonstrate on several benchmarks.

**Summary Of The Review:**

In brief, the paper solves critical problems in AAD through an interesting lightweight, model-agnostic and diversity-aware AAD (LMADA). The nature of model-agnostic allows it to extend to other learning models or tasks with ease. The model has been evaluated extensively on meta-AAD datasets.

---

> ### Author Response · Authors · 2022-11-17
> **Response**
>
> We thank you for the valuable comments. Our response is as below.
>
> Q1: question about efficiency of DPP
>
> In Sec.3.1.2, we use pre-truncation for kernel matrix construction. As we explained in Sec.3.2.1, anomalous samples generally account for a small percentage of the whole dataset compared with the normal class. It is unnecessary to regard all samples as candidates for the sample selector. That is why we can reduce the time complexity of DPP.

---

### Official Review · Reviewer_NYZD · 2022-10-24

**Confidence:** 3
**Correctness:** 2
**Technical Novelty And Significance:** 2
**Empirical Novelty And Significance:** 2
**Recommendation:** 5

**Clarity, Quality, Novelty And Reproducibility:**

The paper is mostly easy to understand, however, some things need better clarification e.g., it is not obvious which base detector has been used for the plots in Figure 5. The paper presents a mildly novel work, but the overall architecture is quite convoluted. It is possible that the presence of the proxy neural network hinders explainability of the anomalies.

**Strength And Weaknesses:**

1. The paper targets an important problem in anomaly detection.

2. Section 4.1 Datasets and Settings: The dataset information is not complete. For example, it is not clear which categories in each dataset were treated as anomaly and normal. The Table 1 in appendix has different number of instances from cited previous works (Das et al., Siddiqui et al.).


3. Section 4.1: "We run 5 feedback iterations and query 20 samples in each iteration. Same" -- 20 samples per iteration is an arbitrary number -- specifically for smaller datasets this is pretty large. (Das et al., Siddiqui et al.) used 1 query per iteration. For fair comparison, need to show results with this setting and another set of ablation experiments where the number of queries per iteration is varied.


4. The paper claims that one of the contributions is the sampling strategy. However, the experiments are not correctly designed to demonstrate its effectiveness. Typically, active learning algorithms have two aspects which are/should be independently replaceable: (a) the query strategy to select samples for user feedback (such as query most anomalous) and, (b) update the algorithm/model parameters with the new labeled data from user. The correct way would be to use the existing benchmark active learning algorithms and replace just their query strategy with the new strategy; then check whether the performance improves/degrades. For the query strategy to be useful and generic, it should work well with other algorithms instead of being tied to a specific one.


5. Section 4.2: "Specifically, we calculate F1-Score on the entire dataset after finishing an iteration of feedback." -- This is inappropriate for active learning. The F1-score should be computed on an independent dataset, or a different metric should be used suitable for active learning.


6. Figure 5: The plot of F1-score along the y-axis is improper. Consider for example if there are exactly 20 true anomalies in the dataset and all get detected in the first iteration. Then F1 will be 1.0 in the first round and then decrease monotonously over successive iterations simply because the precision will decrease. This gives a wrong impression about the algorithm behavior -- that the performance degrades with successive iterations, whereas, that is not true. For an active anomaly detection algorithm, it would be more appropriate to measure the % of true anomalies detected with each feedback iteration (or the AUC).


7. Section 3.2: "In this way, LMADA achieves feedback incorporation in a lightweight manner, only with a non-linear representation transformation." -- It is misleading to call this approach 'lighweight' just because one (last) stage appears to be simple.

**Summary Of The Paper:**

The paper presents a diversity-aware query strategy for active anomaly detection. The proposed query strategy is based on DPP. A positive semi-definite pairwise similarity matrix between highest ranked anomalous instances is first constructed, and then a subset of instances is computed by maximizing its principal minor. The subset of instances selected in this manner has been shown to maximize diversity in previous literature.

**Summary Of The Review:**

The paper leaves out crucial experiments that should compare just the selection strategy across active learning algorithms (i.e., replace the selection strategy in existing benchmark active learning algorithms with the proposed DPP technique) with existing standard strategies. Due to this, the paper falls short of being technically sound.

---

> ### Author Response · Authors · 2022-11-17
> **Response**
>
> We thank you for the valuable comments. Our response is as below.
>
> Q1: questions about datasets statistics
>
> Firstly, only the two datasets, i.e., KDD-SA and KDD-http, have annotated anomalous categories. Other datasets only have binarized labels in ODDS and ADBench, i.e., “normal” and “anomaly”.
>
> Secondly, we have checked the datasets details and confirmed our statistical results are consistent with that on ODDS official website (http://odds.cs.stonybrook.edu/ ), ADBench repo (https://github.com/Minqi824/ADBench ) and sklearn dataset fetch API for KDD-Cup datasets (https://scikit-learn.org/stable/modules/generated/sklearn.datasets.fetch_kddcup99.html ).
> In the previous works (Das et al., Siddiqui et al.), we found that they conducted data sampling. That’s why their data statistics results are different from ours.
>
> For example, in Sec. IV of the paper “Incorporating expert feedback into active anomaly discovery” (Das et al. 2016), they explained that “This implementation requires an n×n kernel  matrix  and therefore  does  not  scale  to  large  datasets.  Therefore, for the larger datasets (Covtype,KDD-Cup-99,Mammography,Shuttle)  we  also  include  results  from  a  smaller  version  of the original dataset which was created by sub-sampling 2000 data instances and keeping the ratio of anomalies to nominals roughly the same as in the original dataset. These sub-sampled datasets are named*-sub (Table I). For the Cardiotocography dataset we retained all instances from the nominal class as in the original dataset, but down-sampled the anomaly instances so that they represent only around 2% of the total data.”
>
> Q2: question about query number per feedback iteration
>
> We conducted the comparison experiment under different query numbers/per feedback iteration (1, ,5, 10, 20) on KDD-SA dataset, which can be found in Fig.12. From the figure, we can see that LMADA can achieve a consistent performance improvement, even with only 1 sample per iteration. On the contrary, the F1-Scores of FIF/GAOD/Meta-AAD fail to increase because they only select the top-ranked samples for updating the model, ignoring the low-ranked anomaly samples, such as the “smurf” type (as we presented in Sec.2.1).
>
> Q3: question about active learning
>
> LMADA is more an end-to-end solution than two separate parts. In our model tuner, described in Sec.3.2.2, we used the repulsion score generated by the sample selector as the weight of consolidation loss. Therefore, we did not separately evaluate the two parts.
>
> Q4: question about evaluation metric
>
> Firstly, we calculate the F1-Score on the remaining unlabeled data, excluding the feedback samples. So, there is no data leakage problem and we do not need another independent dataset for evaluation. In addition, F1-Score is commonly used for evaluating anomaly detectors (Wang et al., 2020, Zhang et al., 2019, Sadaf & Sultana, 2020, John & Naaz, 2019) and thus it is a proper evaluation metric.
>
> Q5: The plot of F1-score along the y-axis is improper……
>
> This is a misunderstanding of our evaluation metric. The F1-Score is calculated on the entire unlabeled dataset instead of the feedback sample set (e.g., 20 samples). Therefore, the model performance would not decrease, but in contrast, increase as more feedback is collected.
>
> Previous AAD work plots the Anomaly Discovery Curve (Siddiqui et al. 2018, Das et al. 2016) which shows the number of discovered anomalies with respect to the number of queries (as in the comments: “% of true anomalies detected with each feedback iteration”). This metric focuses only on precisely capturing true anomalies without false alarms (i.e., precision), but not the miss-detected ones (i.e., recall). As we discuss in Sec.1 and Sec.2.1, recall is also critical for anomaly detection, and therefore we use F1-Score instead of Anomaly Discovery Curve in our paper.
>
> Q6: question about “lightweight”
>
> As we introduced in Sec.2.2, the term “lightweight” in our paper refers to much less parameters of trained LMADA model compared to GAOD and meta-AAD. The latter two AAD approaches leverage neighborhoods of labeled instances to exploit feedback information. They require persisting the entire dataset for neighboring sample retrieval. Therefore, the final tuned detection model would become increasingly heavier and heavier. On the contrary, only the parameters of the non-linear representation adjuster are persisted in our approach and its size will not become larger with continuous arriving data.

---

> > ### Comment · Reviewer_NYZD · 2022-11-22
> > **Thanks**
> >
> > Thanks to the authors for the clarifications.
> >
> > It would have been much more interesting if the query strategy could be separated out and applied with any other active learning algorithm. The tight coupling only narrows the applicability of the paper's contribution. It is still not clear to me why Equations 1, 2, 3 cannot be extended to other active algorithms. Even the definition of the feedback repulsion score from Eqn 3 seems to be only related to samples in a feedback pool and not tied to specific algorithm.
> >
> > The aspect of 'lighweight' is also not entirely satisfactory considering that the additional training and tuning of the various components of Figure 4.
> >
> > However, I am increasing my scores based on the additional clarifications.

---

> > > ### Author Response · Authors · 2022-11-28
> > > **Response**
> > >
> > > Thanks for your reply.
> > >
> > > In this paper, we focus on how to incorporate feedback information in anomaly detection models. Next step, we will study how to generalize our proposed query strategy to more scenarios, beyond the anomaly detection task.
> > >
> > > As we introduced in Sec.2.2, “lightweight” refers to the fewer parameters of the representation adjuster and its size will not become larger with continuous arriving data. Besides, the training and tuning processes in Figure 4 will not influence the inference process of the representation adjuster.
> > >
> > > For example, assume we finished the feedback incorporation procedure and get ready to deploy the updated detection model in production. For LMADA, we only need to insert a well-trained non-linear layer (representation adjuster) between the input sample stream and the original base AD. On the contrary, GAOD and Meta-AAD must persist the entire seen data samples for neighboring sample retrieval (see Sec.2.2), which has a higher overhead than ours in terms of model complexity.

---

> ### Comment · Area_Chair_YAMX · 2022-11-21
> **Any comments to the responses from authors?**
>
> Dear Reviewer NYZD,
>
> Thank you very much for your detailed review.  The authors have provided responses to your questions.  How did they change your evaluation (particularly on the experimental design)?

---

### Official Review · Reviewer_NuRo · 2022-10-26

**Confidence:** 3
**Correctness:** 2
**Technical Novelty And Significance:** 3
**Empirical Novelty And Significance:** 3
**Recommendation:** 6

**Clarity, Quality, Novelty And Reproducibility:**

The novelty and quality of the paper are high. While the problem they are attacking is not novel, the creation of a model-agnostic method by transforming the data rather than the model is novel. The breadth and depth of the empirical results greatly benefit the quality of the proposed method. There are some questions that need to be addressed (see the previous section) to improve the clarity and, possibly, some aspects of the quality of the paper.

**Strength And Weaknesses:**

The paper has strong empirical results, is attacking a significant problem, and has a novel approach to making its method agnostic to the base AD algorithm(s). The performance increases seen by using the proposed method, especially with different types of base AD models, are very convincing of the soundness of the presented method and its underlying insight. I also especially appreciate the way the paper attacks the problem of making an agnostic method by attacking the data featurization as a way of improving the result; it very much embodies the fundamental tenets of data-centric data science.

There are a few weaknesses in (or questions about) the correctness of the proposed method and the clarity of the paper.

•	Why does the method need a proxy model? If the proxy model is frozen at Phase 2 of training when the representation adjuster is trained, why not just take the user feedback and base AD model results and train the neural network adjuster directly? It's not clear to me why there needs to be a proxy model for the method when it seems like one could take the feedback loss and the consolidation loss directly from the base AD outputs to train the representation adjuster.

•	Why is the proxy model a single-layer neural network with no bias term? Roughly speaking, adding depth to a neural network improves its ability to do non-linear transformations, so why not opt for a deeper network for the representation adjuster?

•	How did you decide what the proxy model should be? Why is it a neural network and what is the architecture of that network? Were other models considered?

•	Figure 6 does not really show the comparison of using a base model to using a base model + LMADA, which is the claim in section 4.4. It shows how different base model + LMADA models perform, which is also good, but there needs to be something more to show how adding LMADA for any base model improves that model.


**Summary Of The Paper:**

The paper proposes a method for using any anomaly detection method in an Active Anomaly Detection (AAD) setting. The proposed method consists of three phases: 1) training of a proxy model to emulate the results of the base anomaly detection model(s), 2) training of a data transformation module using human feedback on the anomalous samples, and 3) applying the transformation layer in front of the base anomaly detectors in the data pipeline. The proposed method shows superior results in AAD across several different benchmark datasets. The paper also uses some ablation and comparison results to support the underlying insight beyond the method, namely that there needs to be a diversity-aware sampling strategy for presenting anomalous points to a human.

**Summary Of The Review:**

Given the good insight in the paper (diversity-based sampling) and the solid empirical validation done to support that insight, the paper probably merits publication. However, I do have some concerns about the proposed method that prevents me from being sure that it should be published.

---

> ### Author Response · Authors · 2022-11-17
> **Response**
>
> We thank you for the valuable comments. Our response is as below.
>
> Q1: questions about proxy model
>
> The representation adjuster cannot be trained directly only on top of base AD because the unsupervised detectors (e.g., Isolation Forest) are not gradient-optimizable. As such, the gradient generated by loss cannot be backpropagated to the representation adjuster. As we stated in Sec.3.2.1, we turn unsupervised detectors into gradient-optimizable neural networks, i.e., proxy model, which facilitate the subsequent representation adjuster tuning.
>
> Q2: questions about proxy model architecture
>
> We tried some complicated and deeper network architecture during the evaluation, e.g., transformer and CNN, as the proxy model. However, we found they achieved minor performance gain compared with the single-layer network on the 8 benchmark datasets. It manifests that a single-layer network is sufficient to mimic the behavior of most common unsupervised base detectors.
> Actually, the structure of the proxy model depends more on the modality of input data. For example, CNN model should be more suitable for image anomaly detection task and Transformer more suitable for natural language or sequence data anomaly detection task. We will leave it for future work.
>
> Q3: questions about comparison between base AD and base AD + LMADA
>
> We showed, in Fig.6, the F1-Score of the base AD model. It is the first point of each curve, where x=0, which denotes the performance of base AD without any user feedback.

---

> > ### Comment · Reviewer_NuRo · 2022-11-27
> > **Thank you for the replies**
> >
> > For question Q1, I am still not clear why there needs to be a proxy model. I understand that Base AD detectors can be both unsupervised and not gradient optimizable. However, why can't one just take the loss from the base AD detector and just apply it directly to the representation adjuster? Since the representation adjuster can have any loss function plugged into it, why not use the loss from the base AD detector and use that as the loss to make weight adjustments on the representation adjuster? Put another way, why does the loss have to flow through a proxy model before being applied to the representation adjuster? Does doing so make the tuning more stable perhaps?

---

> > > ### Author Response · Authors · 2022-11-28
> > > **Response**
> > >
> > > Thanks for your reply.
> > >
> > > In LMADA, the training process of the model tuner can be simply summarized as “x_train -> representation adjuster -> proxy model -> loss function (Eq.4)”. After feedback incorporation, we use “x_test -> representation adjuster -> base AD -> anomaly score” for inference. The details are elaborated in Sec.3.2 and Fig.4.
> > >
> > > Firstly, if we replace Eq.4 loss function as the base AD loss (i.e., x_train -> representation adjuster -> base AD -> base AD loss function), the loss of base AD cannot be backpropagated to the representation adjuster because the base ADs discussed in our paper (e.g., IF, LODA) are not gradient-optimizable. Secondly, we directly apply base AD loss to the representation adjuster (i.e., x_train -> representation adjuster -> base AD loss function), the model cannot be trained because the base AD loss is calculated on top of the modeling process of different base ADs and cannot be directly applied to the representation adjuster. Besides, as we stated in Sec.3.2, the model tuner focuses on how to incorporate newly LABELED data points. However, the base AD is UNSUPERVISED, and its loss function cannot consume feedback labels directly.
> > >
> > > During the inference, the sample to be detected is transformed by the representation adjuster and then FED BACK to the base AD to estimate its anomaly score (i.e., x_test -> representation adjuster -> base AD -> anomaly score). Compared with the training phase, base AD, instead of the proxy model, takes the transformed representations as the input. As such, the proxy model is required to approximate the base AD, making the transformed representation vectors can be seamlessly input to the base AD. We also have conducted related experiments in Sec.4.6 and Appendix A.7.

---

> > > > ### Comment · Reviewer_NuRo · 2022-11-28
> > > > **Response to Response**
> > > >
> > > > I believe I understand now. This was the answer I was looking for: "the model tuner focuses on how to incorporate newly LABELED data points. However, the base AD is UNSUPERVISED, and its loss function cannot consume feedback labels directly."

---

> ### Comment · Area_Chair_YAMX · 2022-11-21
> **Any comments to the responses from authors?**
>
> Dear Reviewer NuRo,
>
> Thank you very much for your informative review.  The authors have provided responses to your questions.  Did they clarify what was unclear?  How did they change your evaluation, particularly on the correctness?

---

### Decision · Program_Chairs · 2023-01-20

**Decision:**

Accept: poster

**Justification For Why Not Higher Score:**

Since the novelty of the proposed method is not particularly high, the paper is likely to have impact within the area of active anomaly detection.

**Justification For Why Not Lower Score:**

The paper proposes a novel approach with strong empirical support for the important subject of active anomaly detection.

**Metareview: Summary, Strengths And Weaknesses:**

This paper proposes a new method for active anomaly detection (AAD), where users can give feedback on the results of anomaly detection to tune the anomaly detection model.  The proposed approach overcomes two limitations of existing approaches.  First, while the existing approaches select most anomalous samples to get users' feedback, the proposed approach induces diversity in the samples via determinantal point processes (DPP).  Second, while the base anomaly detection method and the active sampling methods are tightly coupled in existing approaches, the proposed framework allows the decomposition, so that the proposed approach can work with any base anomaly detection method.

A major strength of the paper is in its strong empirical support, which demonstrates that the proposed approach significantly outperform existing AAD methods in a wide range of tasks.  This strength clearly outweigh the paper's weakness that the technical novelty of the proposed approach is rather limited, as the use of DPP to induce diversity in this context is rather standard.

**Note From Pc:**

if the above contains the word "oral" or "spotlight" please see: "oral" presentation means -> notable-top-5% and "spotlight" means -> notable-top-25%. As stated in our emails, we are disassociating presentation type from AC recommendations